# Trastuzumab Rechallenge in HER2-Positive Metastatic Breast Cancer: A Promising Strategy for Enhanced Progression-Free Survival Post Ado-Trastuzumab Emtansine Progression

**DOI:** 10.3390/medicina60122069

**Published:** 2024-12-16

**Authors:** Yunus Emre Altıntaş, Oğuzcan Kınıkoğlu, Anıl Yıldız, Deniz Işık, Uğur Özkerim, Sıla Öksüz, Tuğba Başoğlu Tüylü, Heves Sürmeli, Nedim Turan, Hatice Odabaş

**Affiliations:** 1Department of Medical Oncology, Kartal Dr. Lütfi Kırdar City Hospital, Istanbul 34865, Türkiye; ogokinikoglu@yahoo.com (O.K.); dnz.1984@yahoo.com (D.I.); ugur.ozkerim@hotmail.com (U.Ö.); sila.oksuz@gmail.com (S.Ö.); basoglutugba@gmail.com (T.B.T.); hevessurmeli@hotmail.com (H.S.); turan.nedim@hotmail.com (N.T.); odabashatice@yahoo.com (H.O.); 2Department of Medical Oncology, Istanbul University Oncology Institute, Istanbul 34093, Türkiye; anilyildiz@live.com

**Keywords:** HER2-positive breast cancer, trastuzumab re-challenge, lapatinib capecitabine

## Abstract

*Background and Objectives*: Metastatic breast cancer (MBC), particularly the HER2-positive subtype, represents a significant clinical challenge, with approximately 20–25% of breast cancer cases demonstrating HER2 overexpression. Trastuzumab, a monoclonal antibody targeting HER2, has significantly improved outcomes in these patients. However, progression after second-line treatments such as trastuzumab emtansine (T-DM1) necessitates exploring subsequent therapeutic options. This study aims to compare the efficacy of trastuzumab plus gemcitabine (GT) with lapatinib plus capecitabine (LC) as third-line treatments in HER2-positive MBC post-T-DM1 failure. *Materials and Methods*: This retrospective study included 98 HER2-positive MBC patients treated between 2017 and 2023 who progressed after T-DM1. Patients were divided into two groups: 21 received GT, and 28 received LC. Key endpoints included progression-free survival (PFS), overall survival (OS), objective response rate (ORR), and adverse events. Statistical analyses were performed using SPSS 26.0, with Kaplan–Meier survival curves, log-rank tests, and Cox proportional hazards models. *Results*: Median PFS was significantly longer in the GT group (9.5 months) compared to the LC group (4.3 months, *p* = 0.02). OS was also higher for GT (22.1 months vs. 10.0 months, *p* = 0.02). ORR favored the GT group (33.3% vs. 10.7%, *p* = 0.046), and progressive disease was more common in the LC group (57.1% vs. 33.3%, *p* = 0.022). The safety profile showed higher rates of diarrhea in the LC group, but both regimens were generally well tolerated. *Conclusions*: Trastuzumab re-challenge with gemcitabine demonstrated superior PFS, OS, and ORR compared to lapatinib plus capecitabine, suggesting it may be a more effective third-line therapy in HER2-positive MBC patients who have progressed after T-DM1. Further prospective studies are needed to confirm these findings and optimize treatment sequencing.

## 1. Introduction

Metastatic breast cancer (MBC) remains a significant challenge in oncology, with approximately 30% of patients progressing to metastatic disease despite initial treatment efforts [1]. Human epidermal growth factor receptor 2 (HER2)-positive breast cancer constitutes a distinct subtype characterized by overexpression of the HER2 protein, affecting approximately 20–25% of breast cancer cases and associated with decreased survival [2]. Targeted therapies such as trastuzumab, a monoclonal antibody against HER2, have revolutionized the management of HER2-positive MBC, significantly improving survival outcomes when combined with chemotherapy [3].

Trastuzumab emtansine (T-DM1), an antibody–drug conjugate comprising trastuzumab linked to a cytotoxic agent, emerged as a promising second-line therapy for HER2-positive MBC following progression on trastuzumab-based regimens. The EMILIA trial established the survival benefits of T-DM1 over the lapatinib + capecitabine combination [4]. In this setting, median progression-free survival (PFS) efficacy, resistance, or disease progression inevitably occurs, necessitating subsequent treatment strategies. In such cases, choosing between available options, including continued HER2 blockade with alternative agents or switching to non-HER2-directed therapies, poses a clinical dilemma.

More recently, trastuzumab deruxtecan (T-DXd) has emerged as a promising therapy for HER2-positive MBC patients who have progressed on prior HER2-targeted therapies, including T-DM1 [5]. This novel agent has shown encouraging efficacy in clinical trials, demonstrating significant tumor shrinkage and improved survival outcomes [6]. However, it is important to note that T-DXd may not be viable in all settings. Due to factors such as cost and reimbursement policies, its availability and accessibility can be limited by specific countries’ payment conditions.

Lapatinib, a small-molecule dual tyrosine kinase inhibitor targeting HER2 and epidermal growth factor receptor (EGFR), provides an alternative approach [7]. Among the alternatives, lapatinib, a dual HER2 and EGFR inhibitor, in combination with capecitabine, has demonstrated efficacy in HER2-positive MBC, particularly in patients previously exposed to trastuzumab [8]. Studies have shown that lapatinib plus capecitabine yields a median PFS of approximately 6.2 months [9]. However, comparative data evaluating the efficacy and safety of lapatinib plus capecitabine (median PFS: 6.2 months) versus trastuzumab plus chemotherapy (which can achieve a median PFS of 9.2 months depending on the chemotherapy regimen) as second- and third-line treatments following T-DM1 are limited [9,10].

This article aims to critically compare the available evidence regarding the efficacy, safety, and clinical outcomes associated with trastuzumab plus gemcitabine versus lapatinib plus capecitabine in third-line settings following T-DM1 failure in patients with metastatic HER2-positive breast cancer.

## 2. Materials and Methods

### 2.1. Study Design and Patient Selection

This study involved a retrospective review of the medical records of 98 patients with HER2-positive MBC who received third-line therapy after T-DM1 treatment failed and were followed up with between February 2017 and September 2023. Subsequently, we divided the patients into two groups based on whether they were treated with gemcitabine plus trastuzumab (GT) or lapatinib plus capecitabine (LC) after T-DM1 failed in the second-line setting, which is the preferred second-line therapy in our clinic. Patients who received trastuzumab before the re-challenge had been on treatment for at least six months and had at least stable disease at their 3-month evaluations (Figure 1).

Patients qualified for the study if they were female, at least 18 years old, had an ejection fraction of 50% or greater, and had HER2-positive MBC confirmed through pathology. The assessment of HER2-positive status was conducted using immunohistochemical (IHC) analysis (with a result of 3+ indicating positive status), fluorescence in situ hybridization (where 2+ with an amplification ratio exceeding 2.0 indicated positive status), or a combination of both methods. According to the guidelines from the American Society of Clinical Oncology/College of American Pathologists, tumors were classified as hormone receptor (HR)-positive when the IHC results yielded an Allred score of 3 or higher for both estrogen and progesterone receptors.

Clinical and pathological stages, previous treatments, metastatic sites, specifics of treatment regimens (including dosing, schedule, and duration), and treatment outcomes were gathered from medical records. Patients were administered trastuzumab, starting with a loading dose of 8 mg/kg, followed by 6 mg/kg intravenously every 21 days. Gemcitabine was given in a dose of 1250 mg/m^2^ on days 1 and 8 of the 21-day cycle, administered on the same day after the trastuzumab infusion. Patients received lapatinib in a daily dose of 1250 mg continuously, along with capecitabine at 2000 mg/m^2^ divided into two doses from days 1 to 14 of a 21-day cycle. For the hormone receptor-positive group, an anti-hormonal treatment was provided, which included either letrozole (2.5 mg/day), anastrozole (1 mg/day), exemestane (25 mg/day), or tamoxifen (20 mg/day), based on the physician’s choice. Gonadotropin-releasing hormone analogs were administered in monthly or quarterly doses for patients who were perimenopausal.

### 2.2. Assessments

The main objective of this study was progression-free survival (PFS) and overall survival (OS), while the secondary objective focused on the objective response rate (ORR) and safety. ORR was categorized as a complete response (CR) or a partial response (PR). The disease control rate (DCR) included CR, PR, or stable disease (SD). Treatment responses were evaluated every three months until disease progression, death, or loss of follow-up for patients who discontinued for other reasons, following the Response Evaluation Criteria in Solid Tumors [11]. PFS was determined as the number of months from the initiation of third-line treatment to disease progression or death, whichever occurred first. OS was defined as the number of months from the start of third-line treatment until death. Adverse events were classified during follow-ups based on the common terminology criteria for adverse events, version 3.0 [12].

### 2.3. Statistical Analysis

The survival data analysis estimated 95% of confidential intervals (CIs) using the exact method. Chi-square or Fisher’s exact tests were utilized to evaluate the clinical and pathological features of the patients. A *p*-value less than 0.05 was deemed statistically significant. Kaplan–Meier survival curves along with log-rank tests were applied to examine PFS and OS. Univariate and multivariate Cox proportional hazards models were employed to identify factors linked to PFS and OS. Variables with a *p*-value of less than 0.10 in the univariate analysis and those that might influence survival were incorporated into the multivariate analysis. As this study was retrospective, the sample size was determined by the available cohort of patients meeting the inclusion and exclusion criteria. No formal power calculation was performed prospectively. Missing data were excluded from the survival analysis. A statistical analysis was performed using SPSS Statistics 26.0 (IBM corporation, Armonk, NY, USA). TranslateGPT and Grammarly were used for language translation and grammar refinement in the preparation of this manuscript.

### 2.4. Ethical Statement

This research was conducted following the principles outlined in the Declaration of Helsinki and received approval from the Ethics/Institutional Review Board of Kartal Dr. Lütfi Kırdar City Hospital (date: 26 July 2024, no: 2024/010.99/6/4).

## 3. Results

### 3.1. Study Population and Disease Characteristics

The study cohort retrieved 98 patients with HER2-positive MBC who received T-DM1 and included 49 patients: 21 receiving GT and 28 receiving CL in the third-line setting. The median age of the cohort was 49 years (range: 28–83), with the median ages for the GT and CL groups being 50 years (range: 28–64) and 45 years (range: 31–83), respectively. The median duration of trastuzumab exposure was 14 months (range: 6.1–65.3).

Premenopausal status was observed in 49% of the total cohort, with 33.3% in the GT group and 60.7% in the CL group. Postmenopausal patients constituted 51% of the total cohort, with a notably higher proportion in the GT group (66.7%) compared to the CL group (39.3%) (*p* = 0.09). ECOG performance status (PS) was 0–1 across the entire cohort and was evenly distributed between the groups. Hormone receptor (HR) status showed that 71.4% of the total cohort was HR-positive, with 66.7% in the GT group and 75% in the CL group. HER2 status was evenly distributed between the groups.

Metastasis at diagnosis was similar across both groups, with 44.9% of the cohort diagnosed with metastatic disease at onset.

Prior exposure to pertuzumab was significantly different between the groups, with 36.7% of the total cohort having prior exposure. Specifically, 19.0% of the GT group and 50% of the CL group had previous exposure to pertuzumab, reflecting a statistically significant difference (*p* = 0.04). Disease progression was noted in 91.8% of the total cohort, with 95.2% in the GT group and 89.3% in the CL group experiencing progression (*p* = 0.62). These findings provide a comprehensive overview of the demographic and disease characteristics of the study cohort, highlighting the comparability of the groups in most aspects, except prior pertuzumab exposure and the higher proportion of postmenopausal patients in the GT group (Table 1).

### 3.2. Survival Outcomes

The median duration of follow-up was 19.0 months (range = 2.0–65.7). The entire group had a median PFS of 4.8 months (95% confidence interval [CI], 3.3–6.3) and a median OS of 17.3 months (95% CI, 6.7–27.8).

Patients receiving GT in the third line had a median PFS of 9.5 months (95% CI, 3.2–15.8). In contrast, patients receiving CL had a median PFS of 4.3 months (95% CI, 3.1–5.4) (Figure 2). For OS, patients receiving GT had a median OS of 22.1 months (95% CI, 18.2–25.9), while patients receiving CL had a median OS of 10.0 months (95% CI, 0.0–20.9) (Figure 3). The combination of GT significantly increased median PFS compared to those who received CL. This improvement was observed in both the univariate (9.5 vs. 4.3 months; HR, 0.49; 95% CI, 0.27–0.90; *p* = 0.02) and multivariate (HR, 0.49; 95% CI, 0.27–0.90; *p* = 0.01) analyses (Table 2).

Furthermore, GT was associated with a numerically better median OS in univariate analyses (22.1 months vs. 10.0 months, HR, 0.67; 95% CI, 0.33–1.36; *p* = 0.26) compared to the CL group, and this difference was found to be significant in the multivariate analysis (HR, 0.67; 95% CI, 0.33–1.36; *p* = 0.02) (Table 3).

As mentioned, a higher percentage of patients in the CL group (50%) had previous exposure to pertuzumab than those in the GT group. Therefore, additional analyses were conducted with patients who had not received prior pertuzumab, and it was found that even among patients without prior pertuzumab exposure, the GT group still showed significantly better PFS compared to the CL group (hazard ratio, 0.38; 95% confidence interval, 0.17–0.83, *p* = 0.016).

### 3.3. Tumor Response Outcomes Between GT and CL Treatment Groups

The ORR, CR, PR, SD, progressive disease (PD), and DCR were compared between the two groups. The *p*-value test indicated significant differences in the response rates between the treatment groups. Notably, the ORR for GT was 33.3% (95% CI, 13.3–53.3%) compared to 10.7% (95% CI, 0.0–23.5%) for CL (*p* = 0.046). Additionally, a significant difference in the rates of progressive disease was observed, with a *p*-value of 0.022, indicating a higher rate of progression in the CL group. These results suggest that the GT treatment group may have had more favorable tumor response outcomes than the CL group (Table 4).

These results highlight the differences in tumor responses between the two treatment groups, indicating that GT may have a higher overall response rate and a lower rate of progressive disease compared to CL.

### 3.4. Adverse Events

The most common adverse events in the CL group were diarrhea, anemia, hand–foot syndrome, fatigue, and neutropenia, with diarrhea being significantly more frequent (71.4%, 95% CI, 55.5–87.2%) compared to the GT group (19.0%, 95% CI, 4.9–33.2%; *p* < 0.01). In the GT group, anemia, fatigue, diarrhea, hand–foot syndrome, and cardiovascular disorders were observed, in descending order of frequency. Most adverse events in both groups were grade 1–2, and no grade 3–4 events or fatal side effects were reported. The significant difference in diarrhea rates highlights a notable tolerability issue with the CL regimen, while the GT regimen demonstrated a more favorable safety profile overall (Table 5).

## 4. Discussion

Our study indicates that in patients with HER2-positive MBC previously treated with trastuzumab and progressed on TDM-1, re-challenging with trastuzumab in combination with gemcitabine may result in a longer median PFS compared to CL. Based on limited available data, this finding contributes to the growing interest in trastuzumab re-challenge strategies and offers potential insights for optimizing treatment sequencing in this patient population. The potential synergy between systemic and loco-regional approaches highlights the need for further prospective studies to clarify their respective roles.

The concept of trastuzumab re-challenge in HER2-positive MBC has emerged as a therapeutic strategy due to the limited treatment options available after progression on standard therapies. The mechanisms underlying the potential benefit of trastuzumab re-challenge are not fully elucidated. Tripathy et al. demonstrated that some patients who did not initially respond to trastuzumab could respond to a second trastuzumab-based regimen, indicating the potential for continued benefit from this therapy even after disease progression. Although this study was not specifically designed to evaluate the use of trastuzumab beyond progression, the observed 11% response rate and 22% clinical benefit suggest that trastuzumab may still be effective in this context [13].

Moreover, a meta-analysis of seven randomized trials involving 13,864 women with early-stage HER2-positive breast cancer confirmed that adding trastuzumab to chemotherapy significantly reduces the risk of recurrence and death, with benefits persisting for up to 10 years. The results show that trastuzumab substantially reduces the risk of breast cancer recurrence by 34% and breast cancer mortality by 33% compared to chemotherapy alone. The absolute reduction in the 10-year risk of recurrence was 9.0%, and the reduction in breast cancer mortality was 6.4%, with no significant increase in deaths unrelated to breast cancer [14]. This long-term efficacy highlights the potential of trastuzumab re-challenge to provide sustained clinical benefits.

Additionally, a randomized phase II trial comparing trastuzumab plus capecitabine to lapatinib plus capecitabine in patients with HER2-positive MBC previously treated with trastuzumab and taxanes found no significant differences in PFS (6.1 months vs. 7.1 months; HR, 0.81; 95% CI, 0.55–1.21; *p* = 0.39) and OS (31.0 months vs. NA; HR, 0.58; 95% CI, 0.26–1.31; *p* = 0.18) between the two regimens [15]. This suggests that continuing trastuzumab-based therapy beyond progression may be as effective as switching to alternative HER2-targeted agents.

The mechanisms underlying resistance to trastuzumab often involve alterations in HER2 expression, activation of alternative signaling pathways, and changes in tumor microenvironment, which can be potentially countered through optimized re-challenge strategies. Furthermore, novel HER2-targeted therapies, including antibody–drug conjugates (ADCs) like T-DM1 and T-DXd, provide additional avenues for overcoming resistance, particularly when combined with other targeted therapies or immunotherapies. The clinical utility of trastuzumab re-challenge continues to be validated, reflecting its role in comprehensive treatment plans for patients with HER2-positive MBC facing therapeutic resistance [16]. These ongoing efforts further validate the importance of exploring trastuzumab re-challenge strategies as part of a comprehensive treatment plan.

One possible explanation for the lower efficacy of lapatinib and capecitabine following pertuzumab exposure, compared to trastuzumab and gemcitabine, is the impact of dual HER2 blockade on HER2 receptor availability. Pertuzumab, when added to trastuzumab, reduces the amount of available HER2 receptors on the plasma membrane, limiting the binding of T-DM1 in cancer cells. This HER2 downregulation may not only affect T-DM1 efficacy but also impair the effectiveness of subsequent HER2-targeted therapies such as lapatinib [17].

In contrast, the cytotoxic effects of gemcitabine, which acts independently of HER2 receptor availability, may explain the more favorable outcomes observed with the trastuzumab and gemcitabine combination following pertuzumab exposure. Gemcitabine has been shown to upregulate HER2 expression in breast cancer cells with initially low HER2 levels, thereby enhancing the efficacy of HER2-targeted therapies such as trastuzumab. In studies, treatment with gemcitabine significantly increased HER2 mRNA and protein levels in various breast cancer cell lines, including MCF7 and MDA-MB-231. This upregulation is believed to be mediated by activating the NF-κB signaling pathway. As a result of this increased HER2 expression, the binding of trastuzumab, particularly in its emtansine-conjugated form (T-DM1), to the cancer cells was also enhanced, leading to improved antitumor effects. This suggests that combining gemcitabine with trastuzumab could be a promising strategy to improve treatment outcomes [18]. Additional analysis showed that even among patients who did not receive prior pertuzumab, the GT group still demonstrated significantly better PFS than the CL group, reinforcing the potential efficacy of the GT regimen over the CL regimen, irrespective of prior pertuzumab exposure.

The combination of trastuzumab with a cytotoxic agent like gemcitabine is supported by preclinical and clinical evidence showing synergistic effects in HER2-overexpressing breast cancer. Preclinical studies have demonstrated that combining gemcitabine and trastuzumab produces additive or synergistic antitumor effects in HER2-positive human breast cancer cell lines. Building on these findings, a multicenter phase II trial was conducted to evaluate the safety and efficacy of the gemcitabine/trastuzumab combination in patients with metastatic breast cancer who had received prior treatments [19].

In a retrospective study, researchers evaluated the clinical activity and toxicity of a trastuzumab plus gemcitabine regimen in heavily pretreated HER2-positive MBC patients. The study found that despite the patients’ extensive prior treatments, the combination regimen was effective, demonstrating a notable response rate, time to progression (TTP), and OS with minimal toxicity. The findings suggest that sequential trastuzumab-based chemotherapeutic regimens can achieve significant clinical benefits and prolonged TTP in HER2-positive MBC patients [20]. However, integrating locoregional treatment in specific scenarios, such as for local disease control or symptom management, may enhance overall treatment strategies as well [21].

Our study contributes to the existing literature by suggesting that a re-challenge with trastuzumab combined with gemcitabine may be a viable and potentially effective third-line treatment option for patients who were previously exposed to trastuzumab at least 6 months prior and maintained stable disease for a minimum of 3 months before progressing on T-DM1. Although the observed improvement in PFS compared to lapatinib and capecitabine was statistically significant, the small sample size highlights the need for further investigation in larger, prospective studies. Importantly, the higher objective response rate (ORR) observed in the GT group (33.3% vs. 10.7%) demonstrates the potential of this regimen to provide better tumor control, which is critical for improving patient quality of life and delaying disease progression in a heavily pretreated population. The lower rate of progressive disease in the GT group further supports its efficacy as a third-line option.

From a safety perspective, our analysis revealed notable differences in adverse event profiles between the regimens. The significantly higher incidence of diarrhea in the CL group (71.4% vs. 19.0%) reflects a key tolerability concern, as this adverse event can considerably impair patient quality of life and adherence to treatment. In contrast, the GT regimen exhibited a more favorable safety profile, with predominantly grade 1–2 adverse events and no fatal side effects, making it a more feasible option for patients with HER2-positive MBC, who often face cumulative toxicities from prior lines of therapy.

It is important to acknowledge the limitations of our study, including its retrospective design, which introduces the potential for selection bias and incomplete data. As patient inclusion was based on available records, this may have led to the exclusion of certain cases that did not meet the predefined criteria or had missing data. The relatively small sample size, with 49 patients meeting the inclusion criteria, limits the statistical power of the analysis and may affect the generalizability of the findings. While we observed significant differences in key outcomes, a larger cohort would provide more robust data and enable more definitive conclusions. Additionally, the lack of prospective randomization further underscores the need for future studies to validate these findings in a larger and more controlled setting. Despite these limitations, the significant differences observed in progression-free survival, overall survival, and objective response rates provide valuable insights into the potential role of trastuzumab rechallenge combined with gemcitabine in this patient population.

Additionally, the optimal timing and duration of trastuzumab re-challenge and potential biomarkers for predicting response remain undetermined. Despite these limitations, our findings underscore the necessity for further research to elucidate the role of trastuzumab re-challenge in HER2-positive MBC. Future studies should focus on identifying predictive biomarkers, optimizing treatment combinations, and determining the most appropriate sequencing of therapies to maximize patient outcomes.

## 5. Conclusions

In conclusion, our study suggests that re-challenging with trastuzumab combined with gemcitabine may serve as a viable third-line treatment option for HER2-positive metastatic breast cancer patients who have been previously treated with trastuzumab, maintained stable disease for at least three months, and progressed on T-DM1 after a minimum six-month trastuzumab exposure. The observed improvement in progression-free survival compared to lapatinib and capecitabine supports the potential efficacy of this combination; notably, this method may be a practical and accessible treatment alternative for lower-income countries where access to newer agents such as T-DXd is limited. However, the small sample size of our study underscores the need for further investigation through larger, prospective studies. Future research should explore optimal sequencing strategies, predictive biomarkers, and individualized approaches to enhance the efficacy of trastuzumab-based re-challenge strategy in this patient population.

## Figures and Tables

**Figure 1 medicina-60-02069-f001:**
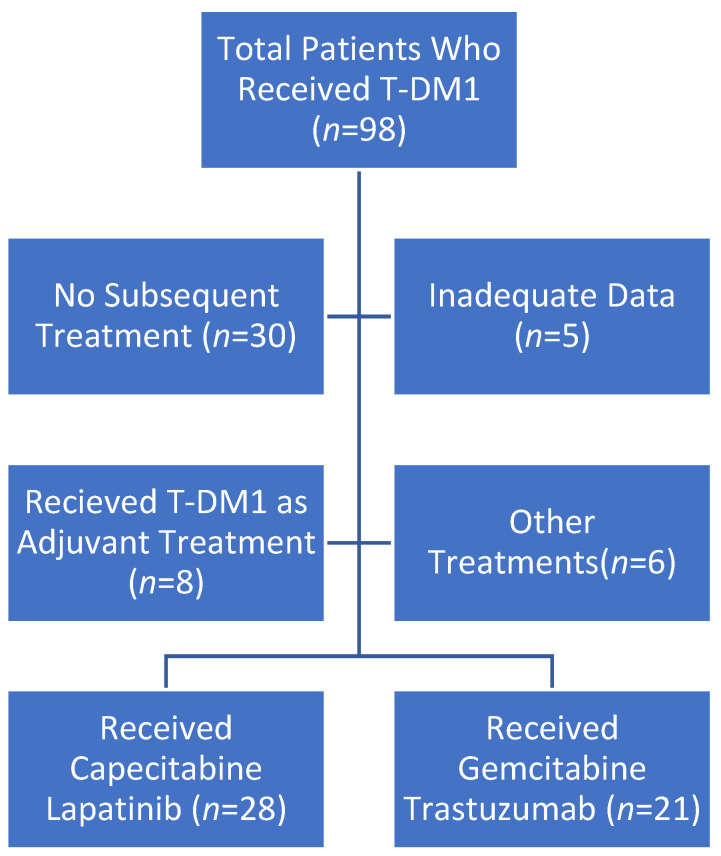
Flow chart describing patients receiving capecitabine–lapatinib or gemcitabine–trastuzumab. Abbreviations: T-DM1, trastuzumab emtansine.

**Figure 2 medicina-60-02069-f002:**
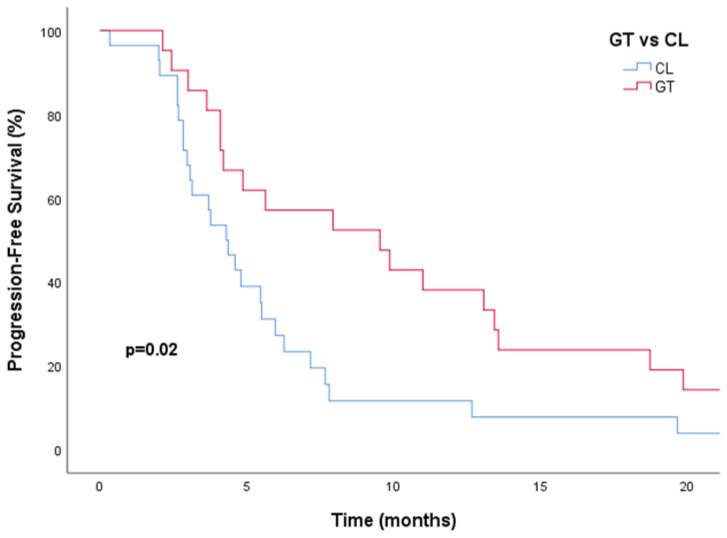
Kaplan–Meier curve indicates a significant improvement in progression-free survival with gemcitabine plus trastuzumab compared to capecitabine plus lapatinib. Abbreviations: CL, capecitabine plus lapatinib; GT, gemcitabine plus trastuzumab.

**Figure 3 medicina-60-02069-f003:**
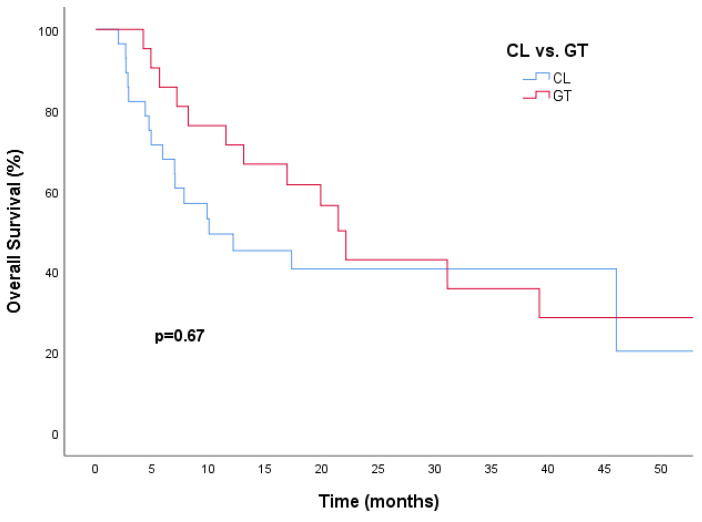
The Kaplan–Meier curve indicates a numerically better OS with gemcitabine plus trastuzumab than capecitabine plus lapatinib. Abbreviations: CL, capecitabine plus lapatinib; GT, gemcitabine plus trastuzumab.

**Table 1 medicina-60-02069-t001:** Comparison of baseline characteristics and treatment outcomes in patients treated with gemcitabine–trastuzumab (GT) vs. capecitabine–lapatinib (CL).

Characteristic	Total Cohort (*n* = 49)	GT (*n* = 21)	CL (*n* = 28)	*p*-Value
Age (years)				
Median (range)	49 (28–83)	50 (28–64)	45 (31–83)	
Menopausal status, *n* (%)				0.09
Premenopause	24 (49)	7 (33.3)	17 (60.7)	
Postmenopause	25 (51)	14 (66.7)	11 (39.3)	
ECOG PS, *n* (%)				0.44
0	42 (85.7)	17 (81.0)	25 (89.3)	
1	7 (14.3)	4 (19.0)	3 (10.7)	
HR status, *n* (%)				0.53
HR-positive	35 (71.4)	14 (66.7)	21 (75)	
HR-negative	14 (28.6)	7 (33.3)	7 (25)	
HER2 status, *n* (%)				0.98
2+ *	16 (32.7)	7 (33.3)	9 (32.1)	
3+	33 (67.3)	14 (66.7)	19 (67.9)	
Denovo metastaticn (%)				0.99
Yes	22 (44.9)	9 (42.9)	13 (46.4)	
No	27 (55.1)	12 (57.1)	15 (53.6)	
Visceral disease involvement, *n* (%)				0.99
Yes	47 (95.9)	21 (100)	26 (92.9)	
No	2 (4.1)	0 (0)	2 (7.1)	
Prior pertuzumab exposure, *n* (%)				0.04
Yes	18 (36.7)	4 (19.0)	14 (50)	
No	31 (63.3)	17 (81.0)	14 (50)	
Progression, *n* (%)				0.62
Yes	45 (91.8)	20 (95.2)	25 (89.3)	
No	4 (8.2)	1 (4.8)	3 (10.7)	
Exitus status, *n* (%)				0.76
Exitus	33 (67.3)	15 (71.4)	18 (64.3)	
Alive	16 (32.7)	6 (28.6)	10 (35.7)	

Abbreviations: ECOG PS; HR, hormone receptor; Eastern Cooperative Oncology Group performance status; HER2, human epidermal growth factor receptor. GT, gemcitabine–trastuzumab; CL, capecitabine–lapatinib. * Fluorescence in situ hybridization with an amplification ratio > 2.0 indicating positive status.

**Table 2 medicina-60-02069-t002:** Univariate and multivariate analysis of factors affecting median progression-free survival in patients treated with gemcitabine–trastuzumab vs. capecitabine–lapatinib.

	Univariate Analysis			Multivariate Analysis	
Variable	mPFS (Months)	HR (95% CI)	*p*-Value	HR (95% CI)	*p*-Value
Age					
<48	5.4	0.85 (0.47–1.55)	0.61	0.88 (0.44–1.77)	0.73
≥48	4.7				
ECOG PS					
0	4.6	0.73 (0.32–1.66)	0.45	0.53 (0.20–1.41)	0.20
1	4.8				
HER2 status					
2+ *	4.2	0.83 (0.43–1.58)	0.57	0.68 (0.33–1.40)	0.30
3+	5.5				
Metastatic at diagnosis					
No	4.7				
Yes	4.8	0.89 (0.48–1.62)	0.71	0.90 (0.43–1.87)	0.79
Prior pertuzumab exposure					
No	5.5	0.98 (0.51–1.86)	0.95	0.66 (0.29–1.51)	0.33
Yes	4.3				
3rd Line Treatment					
GT	9.5	0.49 (0.27–0.90)	0.02	0.38 (0.19–0.80)	0.01
CL	4.3				

Abbreviations: mPFS, median progression-free survival; HR, hazard ratio; ECOG PS, Eastern Cooperative Oncology Group performance status; HER2, human epidermal growth factor receptor. GT, gemcitabine–trastuzumab; CL, capecitabine–lapatinib. * HER2 2+ with fish amplification ratio > 2.0 indicating positive status.

**Table 3 medicina-60-02069-t003:** Univariate and multivariate analysis of factors affecting median overall survival in patients treated with gemcitabine–trastuzumab vs. capecitabine–lapatinib.

	Univariate Analysis			Multivariate Analysis	
Variable	mOS (Months)	HR (95% CI)	*p*-Value	HR (95% CI)	*p*-Value
Age					
<48	31.0	0.52 (0.24–1.09)	0.09	0.49 (0.22–1.13)	0.1
≥48	7.8				
ECOG PS					
0	22.1	0.20 (0.08–0.52)	<0.01	0.17 (0.05–0.54)	<0.01
1	5.6				
HER2 status					
2+ *	12.1				
3+	22.1	0.51 (0.23–1.13)	0.10	0.38 (0.16–0.91)	0.03
Metastatic at diagnosis					
No	22.1	0.77 (0.38–1.56)	0.47	0.70 (0.33–1.48)	0.35
Yes	10.0				
Prior pertuzumab exposure					
No	17.3	0.74 (0.32–1.71)	0.48	0.46 (0.17–1.19)	0.11
Yes	NA				
3rd Line Treatment					
GT	22.1	0.67 (0.33–1.36)	0.26	0.39 (0.17–0.87)	0.02
CL	10.0				

Abbreviations: mOS, median overall survival; HR, hazard ratio; ECOG PS, Eastern Cooperative Oncology Group performance status; HER2, human epidermal growth factor receptor. GT, gemcitabine plus trastuzumab; CL, capecitabine plus lapatinib. * HER2 2+ with fish amplification ratio > 2.0 indicating positive status.

**Table 4 medicina-60-02069-t004:** Tumor response comparison between gemcitabine–trastuzumab and capecitabine–lapatinib treatment groups.

Tumor Response	GT (*n* = 21)	CL (*n* = 28)	*p*-Value
ORR *	7 (33.3)	3 (10.7)	0.046
CR, *n* (%)	0 (0)	0 (0)	NA
PR, *n* (%)	7 (33.3)	3 (10.7)	0.046
SD, *n* (%)	7 (33.3)	9 (32.1)	0.785
PD, *n* (%)	7 (33.3)	16 (57.1)	0.022
DCR †, *n* (%)	14 (66.0)	12 (42.8)	0.197

Abbreviations: GT, gemcitabine–trastuzumab; CL, capecitabine–lapatinib; ORR, objective response rate; CR, complete response; PR, partial response; SD, stable disease; PD, progressive disease; DCR, disease control rate. *, objective response included CR or PR. †, disease control rate was defined as CR, PR, or SD.

**Table 5 medicina-60-02069-t005:** Comparison of adverse events between capecitabine–lapatinib and gemcitabine–trastuzumab.

	CL (28)				GT (21)			
Adverse Event	Grade	No	%	95% CI	No	%	95% CI	*p* of Fisher’s Exact Test
Hand–foot syndrome	1–2	7	33.3	(15.2–51.4)	4	19.04	(4.9–33.2)	0.49
	3–4		0		0	0		NA
Fatigue	1–2	7	33.3	(15.2–51.4)	8	38.09	(18.1–58.0)	0.75
	3–4		0		0	0		NA
Anemia	1–2	8	38.09	(19.5–56.6)	12	57.14	(36.9–77.2)	0.173
	3–4		0		0	0		NA
Diarrhea	1–2	15	71.42	(55.5–87.2)	4	19.04	(4.9–33.2)	<0.01
	3–4	4	19.04		1	4.76		0.337
Neutropenia	1–2	4	19.04	(4.9–33.2)	3	14.28	(0.0–30.0)	0.99
	3–4	0	0		1	4.76		0.445
Cardiovascular disorder	1–2	0	0	(NA)	1	4.76	(4.0–13.9)	NA
	3–4	0	0		0	0		NA

The table compares the incidence and severity of various adverse events between the CL and GT treatment groups, highlighting significant differences in the rates of diarrhea.

## Data Availability

The data presented in this study are available on request from the corresponding author.

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
