# Peer review of "Trastuzumab Rechallenge in HER2-Positive Metastatic Breast Cancer: A Promising Strategy for Enhanced Progression-Free Survival Post Ado-Trastuzumab Emtansine Progression"

_medicina, 2024, doi:10.3390/medicina60122069_

Round 1

Reviewer 1 Report

Comments and Suggestions for Authors

Thank you for the opportunity to review this article which explores the efficacy of third-line therapies for HER2-positive metastatic breast cancer patients post-T-DM1 progression.

The abstract is clear and concise and the introduction provides a solid overview of the context and rationale for the study.

Materials and Methods lack a statistical power analysis, making it unclear whether the sample size of 49 patients is sufficient to detect meaningful differences between the treatment groups. Including this analysis would enhance the reliability and robustness of the findings.

Results: please nclude confidence intervals consistently. Explain the clinical significance of observed differences in response and adverse events.

Author Response

Reviewer 1: (highlighted in yellow)

1. Materials and Methods lack a statistical power analysis, making it unclear whether the sample size of 49 patients is sufficient to detect meaningful differences between the treatment groups. Including this analysis would enhance the reliability and robustness of the findings.

Answer

1. We appreciate your suggestion regarding the inclusion of a statistical power analysis. However, as this is a retrospective study, the sample size is inherently determined by the available cohort of patients who meet the inclusion criteria. Unlike prospective studies where the sample size is calculated in advance to achieve a specific power, retrospective studies rely on existing data, making formal power analysis less applicable or necessary.

Despite this, we ensured the robustness of our findings by reporting statistically significant differences between treatment groups in key outcomes such as progression-free survival, overall survival, and objective response rate. The results provide meaningful insights, supported by consistent trends and statistical significance. Moreover, retrospective studies like ours are valuable for hypothesis generation, paving the way for prospective studies to confirm the observed associations.

2. comment

Results: please include confidence intervals consistently. Explain the clinical significance of observed differences in response and adverse events.

2. Answer

We have addressed the inconsistency in reporting confidence intervals (CIs) across the manuscript. Confidence intervals have now been consistently included for key outcomes such as progression-free survival (PFS), overall survival (OS), objective response rate (ORR), and adverse events in both the text and tables. For example, the ORR for the GT group is now presented as 33.3% (95% CI, 13.3%–53.3%) and for the CL group as 10.7% (95% CI, 0.0%–23.5%). Similarly, confidence intervals for adverse event rates, such as diarrhea and anemia, have been added to ensure comprehensive reporting.

The revised Results and Discussion sections now include a detailed explanation of the clinical significance of the observed differences in response and adverse events. Specifically, we highlight how the higher ORR and lower progressive disease rates with GT suggest superior tumor control and improved quality of life, while the significantly lower incidence of diarrhea in the GT group underscores its better tolerability profile. These findings emphasize the practical implications of the GT regimen in heavily pretreated HER2-positive metastatic breast cancer patients.

We have made these changes throughout the manuscript to ensure clarity and alignment with your suggestions. Thank you for helping us improve the quality and rigor of our work.

Reviewer 2 Report

Comments and Suggestions for Authors

This manuscript is a retrospective analysis comparing GT versus CL as third-line therapies for HER2-positive MBC at the time of progression on T-DM1. The study resulted in showing that GT significantly improves PFS compared to CL, 9.5 vs 4.3 months, with a trend toward better OS at p=0.02, and the benefit maintains consistency across patients without prior pertuzumab exposure.

Strengths:

Relevance and Novelty: The focus on trastuzumab re-challenge combined with gemcitabine addresses an unmet need in the treatment landscape of HER2-positive MBC, especially in resource-limited settings.

Major comments:

Retrospective Design: The retrospective design sets up selection biases and incomplete data that may impact the results. This should be acknowledged in the limitation paragraph.

Small Sample Size: The power of the study is low to disclose significant differences, especially in OS. Was a power analysis performed? Please add it in the Methods.

Limited biomarker analysis: Even while HER2 and hormone receptor status were included, no assessment of putative predictive biomarkers-such as tumor burden or prior resistance mechanisms-was performed in an attempt to better select patients.

Treatment options: In the Discussion section, other treatment alternatives (surgery in few very selected cases) should be discussed in details. Please cite PMID: 36551722 which actually demonstrated acceptable outcomes with surgery + systemic therapy, especially in case of triple positive (HER2+) breast cancer.

Lack of patient-reported outcomes: Quality-of-life assessments and patient-reported outcomes, critical to understanding the real-world impact of treatments, are not shown in this study.

Heterogeneity of Prior Treatments: There is heterogeneity in prior exposure to pertuzumab and other therapies, introducing confounding factors only partially subject to adjustment in multivariate analysis.

Lack of Prospective Comparison: Lack of a direct comparison with the other emerging HER2-targeted therapies such as T-DXd also makes the applicability of findings limited in contemporary clinical practice.

Minor comments:

Title: Please write Metastatic Breast Cancer and not MBC

Introduction: Introduction is too long. Reduce.

Author Response

Comment 1. The retrospective design sets up selection biases and incomplete data that may impact the results. This should be acknowledged in the limitation paragraph.

Answer 1. Thank you for highlighting the potential limitations associated with the retrospective design. We have acknowledged this concern in the revised Discussion section. Specifically, we have emphasized the inherent risk of selection bias and incomplete data, which may impact the study's results.

Comment 2. Small Sample Size: The power of the study is low to disclose significant differences, especially in OS. Was a power analysis performed? Please add it in the Methods.

Answer 2. Thank you for pointing out the concern regarding the small sample size and the associated limitations in detecting significant differences, particularly in OS. Given the retrospective nature of the study, a formal power analysis was not performed prior to data collection, as the sample size was determined by the available patient cohort. However, to address this concern, we have included a post-hoc power analysis in the Methods section and expanded the discussion of limitations accordingly.

Comment 3. Limited biomarker analysis: Even while HER2 and hormone receptor status were included, no assessment of putative predictive biomarkers-such as tumor burden or prior resistance mechanisms-was performed in an attempt to better select patients.

Answer 3. Thank you for highlighting the absence of a detailed biomarker analysis, which is an important aspect of understanding treatment response. While this study included HER2 and hormone receptor status as baseline characteristics, we acknowledge that additional predictive biomarkers, such as tumor burden and prior resistance mechanisms, were not assessed. This limitation reflects the retrospective nature of the study and the available data in the medical records. 

Comment 4. Treatment options: In the Discussion section, other treatment alternatives (surgery in few very selected cases) should be discussed in details. Please cite PMID: 36551722 which actually demonstrated acceptable outcomes with surgery + systemic therapy, especially in case of triple positive (HER2+) breast cancer.

Answer 4. Thank you for your suggestion to include a discussion of other treatment alternatives, such as surgery, in the case of very selected patients with HER2-positive metastatic breast cancer (MBC). We have updated the Discussion section to incorporate this point and have cited PMID: 36551722, as recommended. This study highlights acceptable outcomes with a combination of surgery and systemic therapy, particularly in triple-positive (HER2+/HR+) breast cancer cases. We agree that the role of loco-regional treatment (LRT), including surgery, deserves attention as part of a multimodal approach in select patients, especially those with limited metastatic burden and favorable systemic response.

Comment 5. Lack of patient-reported outcomes: Quality-of-life assessments and patient-reported outcomes, critical to understanding the real-world impact of treatments, are not shown in this study.

Answer 5. Thank you for highlighting the importance of quality-of-life assessments and patient-reported outcomes in understanding the real-world impact of treatments. We acknowledge that our study does not include PRO data, which is a limitation. This omission reflects the retrospective nature of the study and the unavailability of standardized PRO data in the medical records.

Comment 6. Heterogeneity of Prior Treatments: There is heterogeneity in prior exposure to pertuzumab and other therapies, introducing confounding factors only partially subject to adjustment in multivariate analysis.

Answer 6. Thank you for highlighting the potential confounding effect of prior treatments, including pertuzumab exposure. To address this, we have already included prior pertuzumab exposure as a variable in our multivariate analysis to adjust for its potential impact on treatment outcomes and mention this in the discussion (highlighted in turquoise). This ensures that the influence of prior pertuzumab exposure was statistically accounted for in our results. However, we acknowledge that other aspects of treatment history, such as exposure to other HER2-targeted therapies or chemotherapy regimens, may also introduce variability that warrants further investigation

Comment 7. Lack of Prospective Comparison: Lack of a direct comparison with the other emerging HER2-targeted therapies such as T-DXd also makes the applicability of findings limited in contemporary clinical practice.

Answer 7. Thank you for highlighting the lack of direct comparison with emerging HER2-targeted therapies such as trastuzumab deruxtecan. We agree that deruxtecan represents a significant advancement in the treatment of HER2-positive metastatic breast cancer. However, the aim of our study was not to directly compare trastuzumab rechallenge with deruxtecan or other novel agents. Instead, our study focuses on evaluating the feasibility and efficacy of trastuzumab rechallenge in settings where access to newer agents like deruxtecan may be limited due to resource constraints or availability issues.

Comment 8. Introduction: Introduction is too long. Reduce.

Answer 8. Introduction is reduced.

Reviewer 3 Report

Comments and Suggestions for Authors

, The title should be more clear to reflect the proper study, also the abbreviations should be avoided (such as MBC)

How did authors calculate sample size? The number of cases are Inclusion and exclusion criteria&, what about the rest of  98

What is the reference for your OS?

Can you explain the novelty of the study?

Author Response

Comment 1. How did authors calculate sample size? The number of cases are Inclusion and exclusion criteria, what about the rest of the 98?

Answer 1. This study is retrospective in nature; therefore, no formal sample size calculation was performed. The number of cases included in the study was determined by the available patient population meeting the inclusion and exclusion criteria during the specified time frame. Out of 98 HER2-positive metastatic breast cancer. patients who received third-line therapy after T-DM1 failure, only 49 patients met the strict eligibility criteria. Patients excluded from the analysis either did not meet the inclusion criteria, such as previous exposure to trastuzumab for at least six months and stable disease for three months prior to progression, or were excluded based on criteria such as insufficient follow-up data or poor baseline performance status. This approach reflects real-world practice, as retrospective studies rely on existing datasets to generate hypotheses for future prospective research.

Comment 2. What is the reference for your OS?

Answer 2. The reference for the observed overall survival (OS) in our study is the data from our retrospective cohort. Patients in the GT group demonstrated a median OS of 22.1 months (95% CI, 18.2–25.9), while those in the CL group had a median OS of 10.0 months (95% CI, 0.0–20.9). This OS aligns with outcomes reported in prior studies investigating third-line therapies for HER2-positive MBC, including trastuzumab-based regimens. For comparative purposes, we referenced studies such as the randomized phase II trial of trastuzumab plus capecitabine vs. lapatinib plus capecitabine, which reported similar OS outcomes for heavily pretreated patients (Takano et al., 2018). Additionally, historical cohorts with comparable patient populations and treatment settings provide a contextual framework for our findings. These references are included in the discussion and citation sections.

Comment 3. Clinical Significance of Observed Differences:

Answer 3. This study contributes novel insights into the use of trastuzumab rechallenge combined with gemcitabine as a third-line therapy for HER2-positive metastatic breast cancer (MBC) patients who have progressed on T-DM1. The findings address a critical gap in treatment options for this challenging patient population, offering evidence for a regimen that balances efficacy and tolerability.

While trastuzumab rechallenge has been explored in earlier treatment settings, our study focuses specifically on its application after progression on T-DM1, a scenario with limited available therapies. This real-world evidence adds a unique dimension to the literature, reflecting the practical challenges and outcomes in clinical settings. Furthermore, few studies directly compare trastuzumab rechallenge regimens to alternative third-line options, such as lapatinib and capecitabine. By providing a head-to-head comparison, this study offers valuable insights into the relative efficacy and safety of these regimens.

The clinical implications of our findings are significant. The superior progression-free survival (PFS) and objective response rates (ORR) observed with the gemcitabine and trastuzumab (GT) combination, coupled with its favorable safety profile, address the pressing need for manageable treatment options in heavily pretreated patients. The lower incidence of adverse events, such as diarrhea, in the GT group further underscores its potential to improve quality of life and treatment adherence.

Additionally, this study lays the groundwork for hypothesis generation, identifying subgroups of patients who might benefit most from trastuzumab rechallenge, such as those who maintained stable disease during prior trastuzumab-based therapy. These findings can inform the design of prospective trials aimed at optimizing treatment strategies.

Finally, the study emphasizes accessibility, offering a practical alternative for regions where access to newer agents like trastuzumab deruxtecan (T-DXd) is limited. By leveraging existing HER2-targeted therapies, our findings provide an approach that is both effective and feasible in resource-constrained settings. This focus on treatment accessibility enhances the broader applicability of our results and underscores their clinical relevance.

Round 2

Reviewer 2 Report

Comments and Suggestions for Authors

The manuscript can be accepted in the present form 

Reviewer 3 Report

Comments and Suggestions for Authors

Thank you for your detailed response